# Efficient Policy Adaptation with Contrastive Prompt Ensemble for Embodied Agents

**Wonje Choi, Woo Kyung Kim, SeungHyun Kim, Honguk Woo**[*]
Department of Computer Science and Engineering
Sungkyunkwan University
{wjchoi1995, kwk2696, kimsh571, hwoo}@skku.edu

## Abstract

For embodied reinforcement learning (RL) agents interacting with the environment, it is desirable to have rapid policy adaptation to unseen visual observations, but achieving zero-shot adaptation capability is considered as a challenging problem in the RL context. To address the problem, we present a novel contrastive prompt ensemble (CONPE) framework which utilizes a pretrained vision-language model and a set of visual prompts, thus enabling efficient policy learning and adaptation upon a wide range of environmental and physical changes encountered by embodied agents. Specifically, we devise a guided-attention-based ensemble approach with multiple visual prompts on the vision-language model to construct robust state representations. Each prompt is contrastively learned in terms of an individual domain factor that significantly affects the agent's egocentric perception and observation. For a given task, the attention-based ensemble and policy are jointly learned so that the resulting state representations not only generalize to various domains but are also optimized for learning the task. Through experiments, we show that CONPE outperforms other state-of-the-art algorithms for several embodied agent tasks including navigation in AI2THOR, manipulation in egocentric-Metaworld, and autonomous driving in CARLA, while also improving the sample efficiency of policy learning and adaptation.

## 1 Introduction

In the literature of vision-based reinforcement learning (RL), with the advance of unsupervised techniques and large-scale pretrained models for computer vision, the decoupled structure, in which visual encoders are separately trained and used later for policy learning, has gained popularity [1, 2, 3]. This decoupling demonstrates high efficiency in low data regimes with sparse reward signals, compared to end-to-end RL. In this regard, several works on adopting the decoupled structure to embodied agents interacting with the environment were introduced [4, 5], and specifically, pretrained vision models (e.g., ResNet in [6]) or vision-language models (e.g., CLIP in [7, 8]) were exploited for visual state representation encoders. Yet, it is non-trivial to achieve zero-shot adaptation to visual domain changes in the environment with high diversity and non-stationarity, which are inherent for embodied agents. It was rarely investigated how to optimize those popular large-scale pretrained models to ensure the zero-shot capability of embodied agents.

Embodied agents have several environmental and physical properties, such as egocentric camera position, stride length, and illumination, which are *domain factors* making significant changes in agents' perception and observation. In the target (deployment) environment with uncalibrated settings on those domain factors, RL policies relying on pretrained visual encoders remain vulnerable to domain changes.

---

[*]Honguk Woo is the corresponding author.

37th Conference on Neural Information Processing Systems (NeurIPS 2023).

Figure 1 provides an example of egocentric visual domain changes experienced by embodied agents due to different camera positions. When policies learned in the source environment are applied to the target environment, zero-shot performance can be significantly degraded, unless the visual encoder could adapt not only to environmental differences but also to the physical diversity

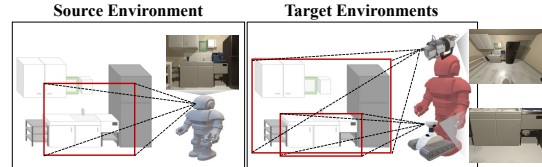

Figure 1: Visual Domain Changes of Embodied Agents

of agents. In this paper, we investigate RL policy adaptation techniques for embodied agents to enable zero-shot adaptation to domain changes, by leveraging prompt-based learning for pretrained models in the decoupled RL structure. To this end, we present CONPE, a novel contrastive prompt ensemble framework that uses the CLIP vision-language model as the visual encoder, and facilitates dynamic adjustments of visual state representations against domain changes through an ensemble of contrastively learned visual prompts. In CONPE, the ensemble employs attention-based state composition on multiple visual embeddings from the same input observation, where each embedding corresponds to a state representation individually prompted for a specific domain factor. Specifically, the cosine similarity between an input observation and its respective prompted embeddings is used to calculate attention weights effectively.

Through experiments, we demonstrate the benefits of our approach. First, RL policies learned via CONPE achieve competitive zero-shot performance upon a wide variety of egocentric visual domain variations for several embodied agent tasks, such as navigation tasks in AI2THOR [9], vision-based robot manipulation tasks in egocentric-Metaworld, and autonomous driving tasks in CARLA [10]. For instance, the policy via CONPE outperforms EmbCLIP [7] in zero-shot performance by 20.7% for unseen target domains in the AI2THOR object navigation. Second, our approach achieves high sample-efficiency in the decoupled RL structure. For instance, CONPE requires less than 50.0% and 16.7% of the samples compared to ATC [1] and 60% and 50% of the samples compared to EmbCLIP to achieve comparable performance in seen and unseen target domains in the AI2THOR object navigation.

In the context of RL, our work is the first to explore policy adaptation using visual prompts for embodied agents, achieving superior zero-shot performance and high sample-efficiency. The main contributions of our work are as follows.

- We present a novel CONPE framework with an ensemble of visual contrastive prompts, which enables zero-shot adaptation for vision-based embodied RL agents.

- We devise visual prompt-based contrastive learning and guided-attention-based prompt ensemble algorithms to represent task-specific information in the CLIP embedding space.

- We experimentally show that policies via CONPE achieve comparable or superior zero-shot performance, compared to other state-of-the-art baselines, for several tasks. We also demonstrate high sample-efficiency in policy learning and adaptation.

- We create the datasets with various visual domains in AI2THOR, egocentric-Metaworld and CARLA, and make them publicly accessible for further research on RL policy adaptation.

## 2 Problem Formulation

In RL formulation, a learning environment is defined as a Markov decision process (MDP) of $(S, A, \mathcal{P}, R)$ with state space $s \in S$, action space $a \in A$, transition probability $\mathcal{P} : S \times A \to S$ and reward function $R : S \times A \to \mathbb{R}$. The objective of RL is to find an optimal policy $\pi^* : S \to A$ maximizing the sum of discounted rewards. For embodied agents, states might not be fully observable, and the environment is represented by a partially observable MDP (POMDP) of a tuple $(S, A, \mathcal{P}, R, \Omega, \mathcal{O})$ with an observation space $o \in \Omega$ and a conditional observation probability [11] $\mathcal{O} : S \times A \to \Omega$.

Given visual domains in the dynamic environment, we consider policy adaptation to find the optimal policy that remains invariant across the domains or is transferable to some target domain, where each domain is represented by a POMDP and domain changes are formulated by different $\mathcal{O}$. We denote domains as $D = (\Omega, \mathcal{O})$. Aiming to enable zero-shot adaptation to various domains, we formulate

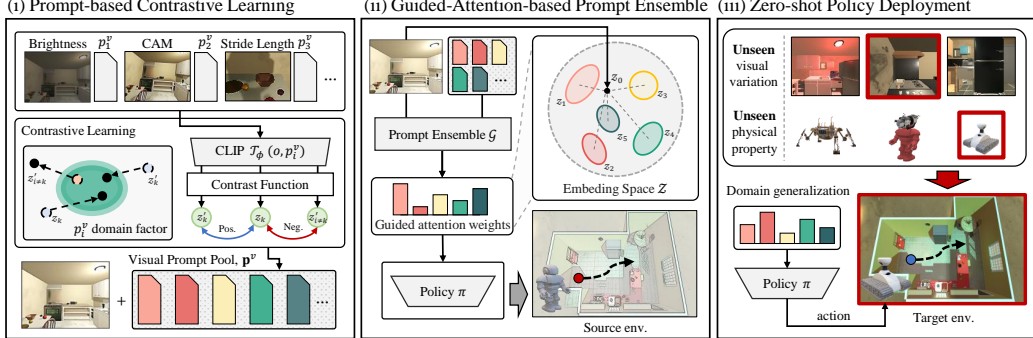

Figure 2: CONPE Framework. The CLIP visual encoder is enhanced offline via (i) prompt-based contrastive learning that generates the visual prompt pool, and a policy is learned online by (ii) guided-attention-based prompt ensemble that uses the prompt pool. In (iii) zero-shot deployment, the policy is immediately evaluated upon domain changes.

the policy adaptation problem as finding the optimal policy $\pi^*$ such that

$$\pi^* = \underset{\pi}{\operatorname{argmax}} \left[ \underset{D \sim p(D)}{\mathbb{E}} \left[ \sum_{t=1}^{\infty} \gamma^t R(s_t, \pi(o_t)) \right] \right] \tag{1}$$

where $p(D)$ is a given domain distribution and $\gamma$ is a discount factor of the environment.

For embodied agents, the same state can be differently observed depending on the configuration of properties such as egocentric camera position, stride length, illumination, and object style. We refer to such a property causing domain changes in the environment as a *domain factor*. Practical scenarios often involve the interplay of multiple domain factors in the environment.

# 3 Our Approach

## 3.1 Framework Structure

To enable zero-shot policy adaptation to unseen domains, we develop the CONPE framework consisting of (i) prompt-based contrastive learning with the CLIP visual encoder, (ii) guided-attention-based prompt ensemble, and (iii) zero-shot policy deployment, as illustrated in Figure 2. The capability of the CLIP visual encoder is enhanced using multiple visual prompts that are contrastively learned on expert demonstrations for several domain factors. This establishes the visual prompt pool in (i). Then, the prompts are used to train the guided-attention-based ensemble with the environment in (ii). To enhance learning efficiency and interpretability of attention weights, we use the cosine similarity of embeddings. The attention module and policy are jointly learned for a specific task so that resulting state representations tend to generalize across various domains and be optimized for task learning. In deployment, a non-stationary environment where its visual domain varies according to the environment conditions and agent physical properties is considered, and the zero-shot performance is evaluated in (iii).

## 3.2 Prompt-based Contrastive Learning

To construct domain-invariant representations with respect to a specific domain factor for egocentric perception data, we adopt several contrastive tasks for visual prompt learning, which can be learned on a few expert demonstrations. For this, we use a visual prompt

$$p^v = [e_1^v, e_2^v, ..., e_u^v], \ e_i^v \in \mathbb{R}^d \tag{2}$$

where $e_i^v$ is a continuous learnable vector with the image patch embedding dimension $d$ (e.g., 768 for CLIP visual encoder) and $u$ is the length of a visual prompt. Let a pretrained model $\mathcal{T}_\phi$ parameterized by $\phi$ maps observations $o \in \Omega$ to the embedding space $\mathcal{Z}$. With a contrast function $P : \Omega \times \Omega \to \{0, 1\}$ [1, 2, 12] to discriminate whether an observation pair is positive or not,

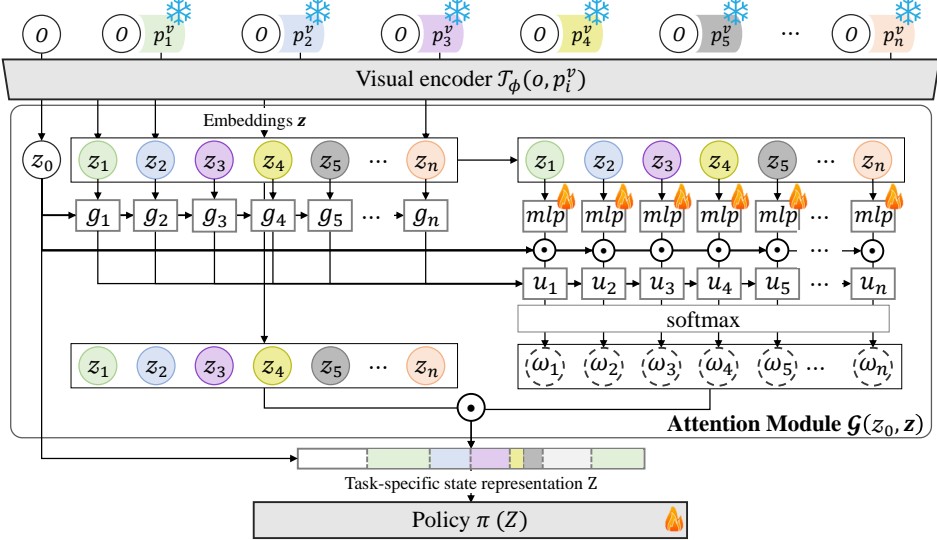

Figure 3: Guided-Attention-based Prompt Ensemble. The cosine similarity-guided attention module $\mathcal{G}$ yields task-specific state representations from multiple prompted embeddings and is learned with a policy network $\pi$.

consider an $m$-sized batch of observation pairs $\mathcal{B}_P = \{(o_i, o_i')\}_{i \le m}$ containing one positive pair $\{(o_k, o_k') | P(o_k, o_k') = 1\}$ for some $k \le m$. Then, we enhance the capability of $\mathcal{T}_\phi$ by learning a visual prompt $p^v$ through contrastive learning, where the contrastive loss function [13] is defined as

$$\mathcal{L}_{\text{CON}}(p^v, \mathcal{B}_P) = -\log\left(\frac{S(\mathcal{T}_\phi(o_k, p^v), \mathcal{T}_\phi(o_k', p^v))}{\sum_{i \ne k} S(\mathcal{T}_\phi(o_i, p^v), \mathcal{T}_\phi(o_i', p^v))}\right), \quad S(x, y) = \frac{1}{\lambda}\exp\left(\frac{\langle x, y\rangle}{\|x\|\|y\|}\right). \quad (3)$$

As in [14], for latent vectors $x$, $y \in \mathcal{Z}$, their similarity in the embedding space $\mathcal{Z}$ is calculated by $S(x, y)$, where $\lambda$ is a hyperparameter. By conducting the prompt-based contrastive learning on $n$ different domain factors, we obtain a visual prompt pool

$$\mathbf{p}^v = [p_1^v, p_2^v, ..., p_n^v]. \quad (4)$$

Through this process, each visual prompt in $\mathbf{p}^v$ encapsulates domain-invariant knowledge pertinent to its respective domain factor.

### 3.3 Guided-Attention-based Prompt Ensemble

To effectively integrate individual prompted embeddings from multiple visual prompts into a task-specific state representation, we devise a guided-attention-based prompt ensemble structure, as shown in Figure 3 where the attention weights on the embeddings are dynamically computed via the attention module $\mathcal{G}$ for each observation.

Given observation $o$ and the learned visual prompt pool $\mathbf{p}^v$, an image embedding $z_0 = \mathcal{T}_\phi(o)$ and prompted embeddings $\mathbf{z} = [z_1 = \mathcal{T}_\phi(o, p_1^v), ..., z_n]$ are calculated. Then, $z_o$ and $\mathbf{z}$ are fed to the attention module $\mathcal{G}$, where attention weights $\omega_i$ for each prompted embedding $z_i$ are optimized. Since directly computing the attention weights using $z_0$ and $\mathbf{z}$ is prone to have an uninterpretable local optima, we introduce a guidance score $g_i$ based on the cosine similarity between the input image and visual prompted image embeddings in $\mathcal{Z}$, i.e., $g_i = \frac{\langle z_0, z_i\rangle}{\|z_0\|\|z_i\|}$. Given that larger $g_i$ signifies a stronger conformity of an observation to the domain factor relevant to the prompted embedding $z_i$, we use $g_i$ to steer the attention weights, aiming to not only improve learning efficiency but also provide interpretability. With guidance $g_i$, we compute the attention weights $\omega_i$ by

$$\omega_i = \frac{\exp(u_i/\tau)}{\sum_k \exp(u_k/\tau)}, \quad u_i = \frac{\langle z_0, k_i\rangle}{\sqrt{d}} g_i \quad (5)$$

**Algorithm 1** Procedure of CONPE Framework

---

Dataset $\mathcal{D} = \{(o_1, o_1'), ...\}$, replay buffer $Z_D \leftarrow \emptyset$, pretrained vision-language model $\mathcal{T}_\phi$
Visual prompt pool $\mathbf{p}^v = [p_1^v, ..., p_n^v]$, attention module $\mathcal{G}$, policy $\pi$

  1: /* *Prompt-based Contrastive Learning* */
  2: **for** $i = 1, ..., n$ **do**
  3:     **while** not converge **do**
  4:         Sample a batch $\mathcal{B}_{P_i} = \{(o_j, o_j')\}_{j \leq m} \sim \mathcal{D}$
  5:         Update prompt $p_i^v \leftarrow p_i^v - \nabla \mathcal{L}_{\mathrm{CON}}(p_i^v, \mathcal{B}_{P_i})$ using (3)
  6:     **end while**
  7: **end for**
  8: /* *Prompt Ensemble-based Policy Learning* */
  9: **for** each environment step **do**
 10:     Sample action $a = \pi(\mathcal{G}(\mathcal{T}_\phi(o), \mathbf{z}))$ using (5), (6)
 11:     $Z_D \leftarrow Z_D \cup \{(\mathbf{z}, a, r)\}$
 12:     Jointly optimize policy $\pi$ and module $\mathcal{G}$ on $\{(\mathbf{z}_j, a_j, r_j)\}_{j \leq m} \sim Z_D$
 13: **end for**

---

where $k_i$ is the projection of $z_i$, $d$ is dimension of $z$, and $\tau$ is a softmax temperature. Then, state embedding $Z$ is obtained by

$$Z = \mathcal{G}(z_0, \mathbf{z}) = z_0 + \sum_{i=1}^{n} \omega_i z_i. \tag{6}$$

Algorithm 1 shows the procedures in CONPE, where the first half corresponds to prompt-based contrastive learning (in Section 3.2) and the other half corresponds to joint learning of a policy $\pi(Z)$ and the attention module $\mathcal{G}$. As $\mathcal{G}$ is optimized by a given RL task objective in the source domains (in line 12), the resulting $Z$ tends to be task-specific, while $Z$ is also domain-invariant by the ensemble of contrastively learned visual prompts based on $\mathcal{G}$ with respect to the combinations of multiple domain factors. The entire algorithm can be found in Appendix.

## 4 Evaluation

**Experiments.** We use AI2THOR [9], Metaworld [15], and CARLA [10] environments, specifically configured for embodied agent tasks with dynamic domain changes. These environments allow us to explore various *domain factors* such as camera settings, stride length, rotation degree, gravity, illuminations, wind speeds, and others. For prompt-based contrastive learning (in Section 3.2), we use a small dataset of expert demonstrations for each domain factor (i.e., 10 episodes per domain factor). For prompt ensemble-based policy learning (in Section 3.3), we use a few source domains randomly generated through combinatorial variations of the seen domain factors (i.e., 4 source domains). In our zero-shot evaluations, we use target domains that can be categorized as either seen or unseen. The seen target domains are those encountered during the phase of prompt-based contrastive learning, while these domains are not present during the phase of prompt ensemble-based policy learning. On the other hand, the unseen target domains refer to those that are entirely new, implying that they are not encountered during either learning phases.

**Baselines.** We implement several baselines for comparison. LUSR [16] is a reconstruction-based domain adaptation method in RL, which uses the variational autoencoder structure for robust representations. CURL [2] and ACT [1] employ contrastive learning in RL frameworks for high sample-efficiency and generalization to visual domains. ACO [12] utilizes augmentation-driven and behavior-driven contrastive tasks in the context of RL. EmbCLIP [7] is a state-of-the-art embodied AI model, which exploits the pretrained CLIP visual encoder for visual state representations.

**Implementation.** We implement CONPE using the CLIP model with ViT-B/32, similar to VPT [17] and CoOp [18]. In prompt-based contrastive learning, we adopt various contrastive learning schemes including augmentation-driven [2, 1, 19] and behavior-driven [12, 20, 21, 22] contrastive learning, where the prompt length sets to be 8. In policy learning, we exploit online learning (i.e., PPO [23]) for AI2THOR and imitation learning (i.e., DAGGER [24]) for egocentric-Metaworld and CARLA.

Table 1: Zero-shot Performance. The policies of each method (CONPE and the baselines) are learned on 4 source domains. The *Source* column presents the performance for those source domains. In all evaluations, we use 30 seen target domains and 10 unseen target domains. The *Seen Target* column presents the performance for the seen target domains, and the *Unseen Target* column presents the performance for the unseen target domains. The unseen target domains are not used for representation learning.

(a) Zero-shot Performance in AI2THOR with Object and Point Goal Navigation Tasks

| Method | ObjectNav. | | | PointNav. | | |
|---|---|---|---|---|---|---|
| | Source | Seen Target | Unseen Target | Source | Seen Target | Unseen Target |
| LUSR | 53.3±1.1 | 21.3±1.9 | 15.1±1.8 | 85.6±4.6 | 71.8±3.8 | 62.4±5.8 |
| CURL | 51.3±1.0 | 8.0±0.1 | 6.9±1.3 | 70.8±7.4 | 55.2±2.7 | 54.8±3.0 |
| ATC | 82.2±9.7 | 72.3±3.3 | 51.3±8.6 | 95.0±3.3 | 89.1±1.9 | 81.9±3.6 |
| ACO | 55.0±23.8 | 39.6±21.5 | 35.8±5.8 | 91.1±6.3 | 73.4±2.0 | 67.5±2.8 |
| EmbCLIP | 89.3±3.0 | 77.6±1.3 | 59.0±6.4 | 95.3±4.6 | 84.5±1.9 | 77.4±1.4 |
| CONPE | **96.3±1.0** | **83.3±0.3** | **79.7±6.4** | **97.8±1.0** | **89.7±1.6** | **84.3±2.0** |

(b) Zero-shot Performance in egocentric-Metaworld with Reach and Reach-wall Tasks

| Method | Reach | | | Reach-Wall | | |
|---|---|---|---|---|---|---|
| | Source | Seen Target | Unseen Target | Source | Seen Target | Unseen Target |
| LUSR | 100.0±0.0 | 46.0±15.1 | 44.7±2.3 | 50.0±10.0 | 33.3±6.1 | 30.7±6.4 |
| CURL | 100.0±0.0 | 53.3±5.0 | 46.7±3.1 | 43.3±15.3 | 2.0±0.0 | 0.7±1.2 |
| ATC | 100.0±0.0 | 71.3±8.1 | 72.0±2.0 | 66.7±5.8 | 5.3±1.2 | 4.0±0.0 |
| ACO | 100.0±0.0 | 52.0±2.0 | 44.0±3.5 | 63.3±15.3 | 8.7±2.3 | 4.7±1.2 |
| EmbCLIP | 100.0±0.0 | 64.7±6.1 | 66.7±4.2 | **100.0±0.0** | 58.0±7.2 | 49.3±5.0 |
| CONPE | 100.0±0.0 | **88.7±3.1** | **86.7±3.1** | **100.0±0.0** | **75.3±3.1** | **67.3±2.3** |

(c) Zero-shot Performance in CARLA with Different Maps

| Method | Map 1 | | | Map 2 | | |
|---|---|---|---|---|---|---|
| | Source | Seen Target | Unseen Target | Source | Seen Target | Unseen Target |
| LUSR | 2141.9 | 635.1±606.2 | 1073.9±212.6 | **2279.6** | 1173.7±914.3 | 2159.4±146.5 |
| CURL | 945.4 | 864.2±638.0 | 1256.0±61.6 | 1050.1 | 1089.9±824.0 | 2190.3±10.2 |
| ATC | 2280.5 | 1684.4±368.2 | 1073.7±618.8 | 2272.2 | 2253.9±218.7 | 2200.1±307.8 |
| ACO | 2265.8 | 1545.6±596.1 | 1330.0±144.5 | 2270.6 | 2360.9±88.0 | 2415.5±53.0 |
| EmbCLIP | 2235.7 | 1732.2±588.6 | 1415.1±669.9 | 2262.7 | 2139.1±655.9 | 2401.3±12.3 |
| CONPE | **2237.5** | **1738.0±163.5** | **1933.4±29.7** | 2277.2 | **2422.5±79.6** | **2512.9±15.7** |

## 4.1 Zero-shot Performance

Table 1 shows zero-shot performance of CONPE and the baselines across source, seen and unseen target domains. We evaluate with 3 different seeds and report the average performance (i.e., task success rate in AI2THOR and egocentric-Metaworld, the sum of rewards in CARLA). As shown in Table 1(a), CONPE outperforms the baselines in the AI2THOR tasks. It particularly surpasses the most competitive baseline, EmbCLIP, by achieving $5.2 \sim 5.7\%$ higher success rate for seen target domains, and $6.9 \sim 20.7\%$ for unseen target domains. For egocentric-Metaworld, as shown in Table 1(b), CONPE demonstrates superior performance with a significant success rate for both seen and unseen target domains, which is $17.3 \sim 24.0\%$ and $18.0 \sim 20.0\%$ higher than EmbCLIP, respectively. For autonomous driving in CARLA, we take into account external environment factors, such as weather conditions and times of day, as domain factors that can influence the driving task. In Table 1(c), CONPE consistently maintains competitive zero-shot performance across all conditions, outperforming the baselines.

In these experiments, LUSR shows relatively low success rates, as the reconstruction-based representation model can abate some task-specific information from observations, which is critical to conduct vision-based complex RL tasks. EmbCLIP shows the most comparative performance among the baselines, but its zero-shot performance for target domains is not comparable to CONPE. In contrast,

CONPE effectively estimates the domain shifts pertaining to each domain factor through the use of guided attention weights, leading to robust performance in both seen and unseen target domains.

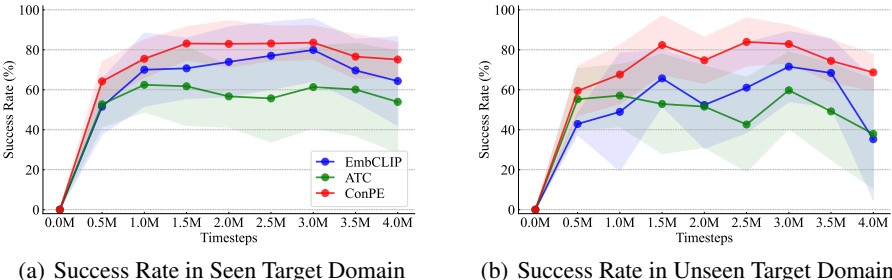

(a) Success Rate in Seen Target Domain      (b) Success Rate in Unseen Target Domain

Figure 4: Sample-efficiency of Prompt Ensemble-based Policy Learning for Object Navigation in AI2THOR. The x-axis represents the number of samples (timesteps) used for policy learning, while the y-axis represents the task success rate for zero-shot evaluation.

**Sample Efficiency.** Figure 4 presents performance with respect to samples (timesteps) that are used by CONPE and baselines for policy learning. Compared to the most competitive baseline EmbCLIP, CONPE requires less than $60.0\%$ timesteps (online samples) for seen target domains and $50.0\%$ for unseen target domains to have comparable success rates.

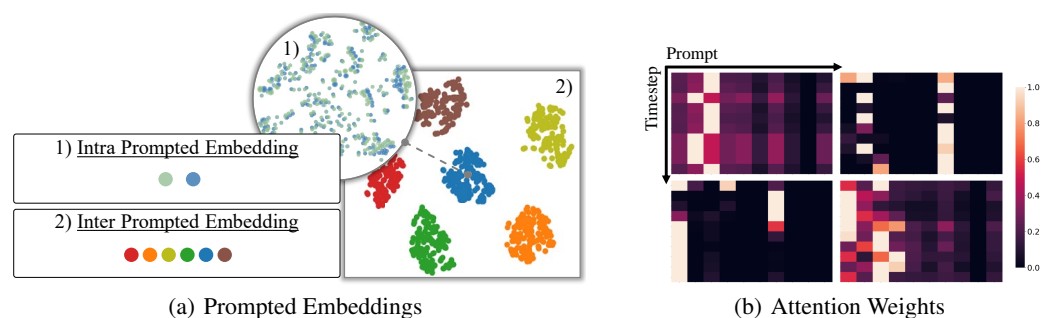

(a) Prompted Embeddings      (b) Attention Weights

Figure 5: Prompt Ensemble Interpretability. In (a), the embeddings in the big circle are intra prompted embeddings obtained by varying domains within a domain factor, and the embeddings in the rectangle are inter prompted embeddings obtained by changing the visual prompts with aligned observation. The closely located intra prompted embeddings indicate the domain-invariant knowledge, while the inter prompted embeddings clustered by different visual prompts indicate the alignment between the visual prompts and the domain factors. In (b), each cell represents attention weight $\omega_i$ applied for prompted embedding $z_i$.

**Prompt Ensemble Interpretability.** Figure 5(a) visualizes the prompted embeddings using the prompt pool obtained through CONPE. For *intra* prompted embeddings, we use observation pairs, where each pair is generated by varying domains within a domain factor. We observe that the embeddings are paired to form the domain-invariant knowledge because the visual prompt is learned through prompt-based contrastive learning. The *inter* prompted embeddings specify that each prompt distinctly clusters prompted embeddings that correspond to the domains associated to its domain factor. Figure 5(b) shows examples of the attention weight matrix of CONPE for four different domains. The x-axis denotes the visual prompts and the y-axis denotes timesteps. This shows the consistency of the attention weights on the prompts across the timesteps in the same domain.

## 4.2 Prompt Ensemble with a Pretrained Policy

While we previously presented joint learning of a policy and the attention module $\mathcal{G}$, here we also present how to update $\mathcal{G}$ for a pretrained policy $\pi$ to make the policy adaptable to domain changes. In this case, we add a *policy prompt* $p_{\text{pol}}^v$ to concentrate on task-relevant features from observations for

the pretrained policy $\pi$ so that prompted embedding $\tilde{z}_0$ with the task-relevant features is incorporated into the guided-attention-based ensemble, i.e., $\pi(\mathcal{G}(\tilde{z}_0, \mathbf{z}))$, where $\tilde{z}_0 = \mathcal{T}_\phi(o, p_{\text{pol}}^v)$.

Table 2: Prompt Ensemble with a Pretrained Policy. The pretrained policies of each task are learned on 4 source domains. The *Source* column presents the performance for those source domains. In all evaluations, we use 40 unseen target domains. The *Target* column presents the performance for the unseen target domains not used for policy training.

(a) Zero-shot Performance in AI2THOR with Visual Navigation and Room Rearrangement Tasks

| Method | ObjectNav. (Aln.) | | PointNav. (Not Aln.) | | ImageNav. (Not Aln.) | | RoomR. (Not Aln.) | |
|---|---|---|---|---|---|---|---|---|
| | Source | Target | Source | Target | Souce | Target | Scoure | Target |
| Pretrained | 87.5±17.2 | 65.8±19.1 | 95.3±4.6 | 80.9±1.6 | 77.2±3.3 | 56.2±2.2 | 87.3±3.1 | 75.2±13.2 |
| CONPE | 88.4±1.7 | **72.8±3.1** | 98.9±1.0 | **84.4±1.0** | 79.2±1.4 | **61.6±1.1** | 93.3±1.2 | **82.2±14.4** |

(b) Zero-shot Performance in egocentric-Metaworld with 4 Different Robot Manipulation Tasks

| Method | Reach (Aln.) | | Reach-Wall (Not Aln.) | | Button-Press (Not Aln.) | | Door-Open (Not Aln.) | |
|---|---|---|---|---|---|---|---|---|
| | Source | Target | Source | Target | Source | Target | Source | Target |
| Pretrained | 100.0±0.0 | 65.7±6.4 | 100.0±0.0 | 58.0±5.8 | 100.0±0.0 | 16.8±2.3 | 100.0±0.0 | 35.6±6.2 |
| CONPE | 100.0±0.0 | **74.7±5.0** | 100.0±0.0 | **75.7±9.0** | 100.0±0.0 | **73.7±8.3** | 100.0±0.0 | **93.2±1.1** |

Table 2 reports zero-shot performance for the scenarios when a pretrained policy is given. We evaluate two different cases: *aligned* (Aln.) when prompt-based contrastive learning is conducted on data from the same task of a pretrained policy; otherwise, *not aligned* (Not Aln.). In AI2THOR, we use data from the object goal navigation task for prompt-based contrastive learning, while each pretrained policy is learned individually through one of tasks including object goal navigation, point goal navigation, image goal navigation, and room rearrangement. Similarly, in egocentric-Metaworld, we use data from the reach task for prompt-based contrastive learning, while each pretrained policy is learned individually through one of tasks including reach, reach-wall, button-press, and door-open. In Table 2(a), CONPE enhances zero-shot performance of the pretrained policies by $3.5 \sim 7.0\%$ for unseen target domains in AI2THOR. This prompt ensemble adaptation requires only 400K samples, equivalent to $10\%$ of the total samples used for policy learning. In Table 2(b), CONPE significantly boosts zero-shot performance of the pretrained policies by $9.0 \sim 57.6\%$ in egocentric-Metaworld.

## 4.3 Ablation Study

Here we conduct ablation studies with AI2THOR. All the performances are reported in success rates.

Table 3: Prompt Ensemble Scalability

| $n$ | Source | Seen Target | Unseen Target |
|---|---|---|---|
| 2 | 98.7±0.4 | 40.5±2.2 | 43.0±2.9 |
| 5 | 96.1±0.6 | 59.2±9.6 | 45.0±10.1 |
| 10 | 96.3±1.0 | 83.3±0.3 | 79.7±6.4 |
| 16 | 91.8±2.0 | 83.8±1.3 | 77.1±6.2 |
| 18 | 98.5±1.8 | 83.3±2.3 | 79.0±4.5 |

Table 4: Prompt Ensemble Methods

| Ensemble Method | Source | Seen Target | Unseen Target |
|---|---|---|---|
| COM-UNI-AVG | 52.9±12.2 | 51.4±7.6 | 43.1±12.7 |
| COM-WEI-AVG | 63.6±11.2 | 42.3±5.2 | 50.3±6.1 |
| ENS-UNI-AVG | 88.6±1.4 | 79.8±3.4 | 65.5±5.0 |
| ENS-WEI-AVG | 94.1±6.3 | 75.5±3.5 | 61.2±12.7 |
| CONPE | **96.3±1.0** | **83.3±0.3** | **79.7±6.4** |

**Prompt Ensemble Scalability.** Table 3 evaluates CONPE with respect to the number of prompts $(n)$. CONPE effectively enhances zero-shot performance for both seen and unseen target domains through prompt ensemble that captures various domain factors. Compared to the case of $n = 2$, for $n = 10$, there was a significant improvement in zero-shot performance for both seen and unseen target domains, with increases of $42.8\%$ and $36.7\%$, respectively. For $n \geq 10$, we observe stable performance that specifies that CONPE can scale for combining multiple prompts to some extent.

**Prompt Ensemble Methods.** Tabel 4 compares the performance of various prompt integration methods [25, 26, 27] including our guided attention-based prompt ensemble. We denote prompt-level integration as COM, and prompted embeddings-level integration as ENS. UNI-AVG and WEI-AVG refer to uniform average and weighted average mechanisms, respectively. CONPE achieves superior success rates over the most competitive ensemble method ENS-UNI-AVG, showing $3.5\%$ and $14.2\%$ performance gain for seen and unseen target domains.

| Table 5: Prompt Ensemble Adaptation | | |
|---|---|---|
| Optimization | Source | Target |
| Pretrained | 95.3±4.6 | 80.9±1.6 |
| w/o $p_{pol}^v$ | 59.5±2.9 | 54.2±0.6 |
| w $p_{pol}^v$ | **98.9±1.0** | **84.4±1.0** |
| E2E | 96.4±0.5 | 83.3±3.3 |

Table 6: Semantic Regularized Data Augmentation

| $\delta$ | w/o Semantic | | w Semantic | |
|---|---|---|---|---|
| | Source | Target | Source | Target |
| 0.1 | 97.4±3.8 | 83.6±8.9 | **100.0±0.0** | **84.1±10.2** |
| 0.2 | 94.7±0.0 | 77.6±12.8 | **94.8±7.4** | **79.7±9.1** |
| 0.3 | 84.2±3.7 | 75.3±8.8 | **96.1±1.9** | **83.1±10.0** |
| 0.4 | 80.3±16.2 | 74.0±14.9 | **86.9±3.8** | **81.3±12.3** |

**Prompt Ensemble Adaptation Method.** Table 5 shows the effect of our ensemble adaptation method for the situation when a pretrained policy is given. As explained in Section 4.2, in this situation, CONPE can update the attention module with an additional prompt $p_{pol}^v$. Note that $p_{pol}^v$ corresponds to this case, while w/o $p_{pol}^v$ corresponds to the other case of using the attention module without $p_{pol}^v$. In addition, E2E denotes the fine-tuning of both the policy and the attention module along with $p_{pol}^v$. The results demonstrate that our method enhances the zero-shot performance of the pretrained policy, showing that $p_{pol}^v$ facilitates the extraction of task-specific features.

**Semantic Regularized Data Augmentation.** So far, we have only utilized vision data, but here, we discuss one extension of CONPE using semantic information. Specifically, we use a few samples of object-level text descriptions to regularize the data augmentation process in policy learning. This aims to mitigate overfitting issues [28, 29]. The detailed explanations can be found in Appendix. As shown in Table 6, CONPE with semantic data (w Semantic) consistently yields better performance than CONPE without semantic data (w/o Semantic) for all noise scale settings ($\delta$). Note that the noise scale manages the variance of augmented prompted embeddings. This experiment indicates that CONPE can be improved by incorporating semantic information.

## 5  Related Work

**Adaptation in Embodied AI.** In the literature of robotics, numerous studies focused on developing generalized visual encoders for robotic agents across various domains [30, 31], exploiting pretrained visual encoders [32, 33], and establishing robust control policies with domain randomized techniques [34, 35]. Furthermore, in the field of learning embodied agents, a few works addressed adaptation issues of agents to unseen scenarios in complex environments, using data augmentation techniques [36, 37, 38, 39, 40] or adopting self-supervised learning schemes [41, 42, 43]. Recently, several works showed the feasibility and benefits of adopting large-scale pretrained vision-language models for embodied agents [7, 44, 45, 46]. Our work is in the same vein of these prior works of embodied agents, but unlike them, we explore visual prompt learning and ensembles, aiming to enhance both zero-shot performance and sample-efficiency.

**Decoupled RL Structure.** The decoupled structure, where a state representation model is separated from RL, has been investigated in vision-based RL [47, 2, 16]. Recently, contrastive representation learning on expert trajectories gains much interest, as it allows expert behavior patterns to be incorporated into the state encoder even when a policy is not jointly learned [1, 20]. They established generalized state representations, yet in that direction, sample-efficiency issues in both representation learning and policy learning remain unexplored.

**Prompt-based Learning.** Prompt-based learning or prompt tuning is a parameter-efficient optimization method for large pretrained models. Prompt tuning was used for computer vision tasks, optimizing a few learnable vectors in the text encoder [18], and it was also adopted for vision transformer models to handle a wide range of downstream tasks [17]. Recently, visual prompting [48] was introduced, and both visual and text prompt tuning were explored together in the multi-modal embedding space [49, 50]. We also use visual prompt tuning, but we concentrate on the ensemble of multiple prompts to tackle complex embodied RL tasks. We take advantage of the fact that the generalized representation capability of different prompts can vary depending on a given task and domain, and thus we strategically utilize them to enable zero-shot adaptation of RL policies.

# 6  Conclusion

**Limitation.** Our CONPE framework exploits visual inputs and their relevant domain factors for policy adaptation. For environments where domain changes extend beyond those domain factors, the adaptability of the framework might be constrained. In our future work, we will adapt the framework with semantic knowledge based on pretrained language models to improve the policy generalization capability for embodied agents in dynamic complex environments and to cover various scenarios associated with multi-modal agent interfaces.

**Conclusion.** In this work, we presented the CONPE framework, a novel approach that allows embodied RL agents to adapt in a zero-shot manner across diverse visual domains, exploring the ensemble structure that incorporates multiple contrastive visual prompts. The ensemble facilitates domain-invariant and task-specific state representations, thus enabling the agents to generalize to visual variations influenced by specific domain factors. Through various experiments, we demonstrated that the framework can enhance policy adaptation across various domains for vision-based object navigation, rearrangement, manipulation tasks as well as autonomous driving tasks.

# 7  Acknowledgement

We would like to thank anonymous reviewers for their valuable comments and suggestions. This work was supported by Institute of Information & communications Technology Planning & Evaluation (IITP) grant funded by the Korea government (MSIT) (No. 2022-0-01045, 2022-0-00043, 2020-0-01821, 2019-0-00421) and by the National Research Foundation of Korea (NRF) grant funded by the MSIT (No. NRF-2020M3C1C2A01080819, RS-2023-00213118).

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
