# OpenReview forum: "Efficient Policy Adaptation with Contrastive Prompt Ensemble for Embodied Agents"
_NeurIPS.cc/2023/Conference — NeurIPS 2023 poster_

### Official Review · Reviewer_FnQE · 2023-07-04

**Soundness:** 3 good
**Presentation:** 2 fair
**Contribution:** 3 good
**Rating:** 6
**Confidence:** 4

**Summary:**

This paper proposes a novel framework with an ensemble of visual contrastive prompts for efficient adaptation of vision-based robot learning tasks, named CONPE. The visual prompts are firstly learned from a contrastive learning procedure with different domain factors, which are expected to be domain orthogonal. Then the prompts are applied for downstream vision robot learning tasks along with a guided-attention-based prompt ensemble module, which adaptively learns an ensemble of vision features and prompts, to obtain task specific and domain invariant visual features, for efficient robot learning tasks. The experimental results demonstrate the effectiveness of the proposed framework.

**Strengths:**

This paper proposes an intuitive yet efficient method to address a meaningful issue of environment generalization in robotics learning. The approach combines the natural of contrastive learning and visual prompt and surpasses existing baselines according to the experimental results.

**Weaknesses:**

1. As shown in Fig. 2 (iii), it indicates that the framework enables robot learning model to generalize to unseen physical property. But according to the statement, the physical property is more likely to refer to angle of view (also can be seen in Fig. 1. However, for most manipulation tasks, the change of the appearance of robot arm/hand brings more disruption, which worths discussion but is missing in this paper.
2. Visual representations in virtual environments are simpler than in real world, which has more complex object layout, texture, etc. Thus, many methods in work in Venv but fail to be deployed. Supportive real-world experimental result is crucial.
3.	Some related works on data-driven environment generalization for robot learning should be discussed, e.g., "Open-World Object Manipulation using Pre-trained Vision-Language Models", and "Pave the Way to Grasp Anything: Transferring Foundation Models for Universal Pick-Place Robots".


**Questions:**

1.	What’s Stage 1’s training data and strategy? How about tune the visual prompts for downstream data in Stage 2?
2.	Several numbers reported in Section 4 worths discussing, e.g., the performance of ATC and EmbCLIP in Tab. 1 (a), why would the methods’ performance on unseen domain be better than seen domain?


**Limitations:**

Environment generalization and adaptation is necessary for robot learning, it is encouraged to explore the potential of visual prompt for generalization of robot arms and other visual factors.

---

> ### Author Rebuttal · Authors · 2023-08-09
>
> ### **Author response for Weakness 1. [physical property]**
> We appreciate the reviewer for this observation. The physical properties that we focus on include camera fields of view, camera positions and action magnitudes which are frequently considered embodiment configurations in the embodied AI literature [1, 2].
> The reviewer's point regarding the impact of the change in the appearance of a robot arm/hand is quite valid. Manipulation tasks can be sensitive to the appearance and configuration of the robot's end effector.
>
> However, given the constraints of the simulators that we have utilized, our concentration on variations of camera fields of view, camera positions, and action magnitudes seemed the most pragmatic and relevant for us. Furthermore, in our current research, our primary emphasis is not on individual domain factors, but rather on our unified framework architecture with visual prompt ensemble, which can address a broad spectrum of adaptation issues.
>
> In future research, we intend to investigate the effects of changes in the robot's appearance on its learning capability and performance.
>
> ### **Author response for Weakness 2. [real-world experimental]**
> We agree that the distinction between virtual environments and real-world scenarios is a valid concern.
>
> The simulator that we used, such as AI2THOR and CARLA is widely recognized for high fidelity and photo-realistic qualities [3, 4, 5].
> While the simulators we employed offer detailed simulations, it is essential to approach the potential of our ConPE framework in real-world applications with caution. We understand that there are inherent differences between these virtual environments and actual real-world conditions. We plan to further explore research utilizing real-world data and applications.
>
> ### **Author response for Weakness 3. [related works]**
> These works [6, 7] mentioned by the reviewer resonate with our goal of harnessing pre-trained models to enhance RL policy adaptation in various areas including robotic manipulation.
>
> We will include these references in the Related Work Section in our revised manuscript. Intriguingly, these works primarily focus on exploring language-grounded policies, providing us with some rational to consider the incorporation of both visual and language encoders.
>
> ### **Author response for Question 1. [training data and strategy]**
> For prompt learning, we use the dataset gathered from various domains, where each domain is determined by varying a specific domain factor while keeping the others consistent.
> We devise rule-based policies for each environment and collect 28,464 samples for AI2THOR, 4,700 samples for Metaworld, and 73,900 samples for CARLA.
>
> In Stage 1 (prompt-based contrastive learning, as depicted in the left-side of Figure 2), given the dataset, we individually train the prompts through several contrastive learning algorithms.
> We describe the details in Appendix A, including behavior-driven contrastive learning, augmentation-driven contrastive learning, and timestep-driven contrastive learning.
>
> In Stage 2 (guided-attention-based prompt ensemble, as depicted in the middle of Figure 2), we train the ensemble module and policy on the source domains, while keeping the visual prompts frozen (which are learned in Stage 1).
>
> ### **Author response for Question 2. [performance scores]**
> In certain cases, the zero-shot performance in unseen target domains can exceed that of the source or seen target domains. While this might appear less consistent, similar observations have been reported in prior works [8, 9] in domain adaptation research.
> This phenomenon can be largely attributed to the disparities between the source and target domains, particularly when the source domains are relatively more challenging than the target domains. We frequently experience this pattern as domains differ by the complex interplay of multiple factors.
>
> ### **Author response for Limitations [generalization of robot arms and other visual factors]**
> We will include the limitation of our work, partially related to domain factors considered in our current experiments, in the Conclusion Section of our revised manuscript.
>
> We appreciate the reviewer's insightful comment on the limitations and agree that further work is needed to explore the potential of our framework toward more complex, realistic situations that are characterized by a multitude of different domain factors and associated with long-horizon control tasks in robotics environments.
>
> [1] Chattopadhyay, Prithvijit, et al. "Robustnav: Towards benchmarking robustness in embodied navigation." Proceedings of the IEEE/CVF International Conference on Computer Vision. 2021.
>
> [2] Shah, Dhruv, et al. "Gnm: A general navigation model to drive any robot." 2023 IEEE International Conference on Robotics and Automation (ICRA). IEEE, 2023.
>
> [3] Duan, Jiafei, et al. "A survey of embodied ai: From simulators to research tasks." IEEE Transactions on Emerging Topics in Computational Intelligence 6.2 (2022): 230-244.
>
> [4] Kolve, Eric, et al. "Ai2-thor: An interactive 3d environment for visual ai." arXiv preprint arXiv:1712.05474 (2017).
>
> [5] Dosovitskiy, Alexey, et al. "CARLA: An open urban driving simulator." Conference on robot learning. PMLR, 2017.
>
> [6] Stone, Austin, et al. "Open-world object manipulation using pre-trained vision-language models." arXiv preprint arXiv:2303.00905 (2023).
>
> [7] Yang, Jiange, et al. "Pave the Way to Grasp Anything: Transferring Foundation Models for Universal Pick-Place Robots." arXiv preprint arXiv:2306.05716 (2023).
>
> [8] Xing, Jinwei, et al. "Domain adaptation in reinforcement learning via latent unified state representation." Proceedings of the AAAI Conference on Artificial Intelligence. Vol. 35. No. 12. 2021.
>
> [9] Li, Dongfen, et al. "Domain adaptive state representation alignment for reinforcement learning." Information Sciences 609 (2022): 1353-1368.

---

### Official Review · Reviewer_gM92 · 2023-07-08

**Soundness:** 3 good
**Presentation:** 3 good
**Contribution:** 3 good
**Rating:** 6
**Confidence:** 3

**Summary:**

This paper proposes a novel contrastive prompt ensemble (ConPE) framework for embodied reinforcement learning (RL) agents. ConPE utilizes a pretrained vision-language model and a set of visual prompts to construct robust state representations. Each prompt is contrastively learned in terms of an individual domain factor that significantly affects the agent's egocentric perception and observation. The attention-based ensemble and policy are jointly learned so that the resulting state representations not only generalize to various domains but are also optimized for learning the task. Through experiments, ConPE outperforms other state-of-the-art algorithms for several embodied agent tasks, including navigation in AI2THOR, manipulation in Metaworld, and autonomous driving in CARLA.

**Strengths:**

This paper presents a novel idea that uses visual prompt-based contrastive learning and guided-attention-based prompt
72 ensemble algorithms for better representation learning that leads to improved vision-based RL policy performance. The idea is neat and the method is technically sound.

The experimental results show that the proposed method can outperform prior vision-based RL methods in navigation and manipulation tasks. The authors also provide extensive ablation studies for better understanding of the algorithm.

**Weaknesses:**

The evaluated tasks in the paper are relatively toy-like. For example, the authors only evaluated the method in reaching tasks for manipulation, which is overly simple. It would be more convincing if the authors could include more challenging and realistic tasks such as more dexterous manipulation tasks like pick and place and kitchen rearrangement and navigation tasks such as Habitat object navigation.

**Questions:**

Clarify and address the concern mentioned in the above section.

**Limitations:**

Yes.

---

> ### Author Rebuttal · Authors · 2023-08-09
>
> ### **Author response for Weakness 1. [more challenging and realistic tasks]**
>
> The simulators that we used (e.g., AI2THOR, CARLA) are widely used in the field of Embodied AI [1, 2, 3, 4, 5].
> The observation state from these environments contains a wide variety of objects, and their image quality is regarded as being photo-realistic [1, 2, 5].
> While we hold the belief that the complexity of the environments we utilized stands strong in comparison to other existing RL environments (e.g., mujoco [6], vizdoom [7] and Megaverse [8]), we agree that there is room to extend our research into more challenging and realistic environments in future work.
>
> Additionally, we believe we have tackled a challenging issue.
> The complexities of practical scenarios often involve the interplay of multiple domain factors, each contributing to the environment overall context in which a policy operates.
>
> Generalizing with respect to each individual domain factor can provide a basic understanding, but we presume that that doesn't necessarily prepare an agent for intricate multi-factorial situations.
>
> For example, consider the car driving example. Mastering driving on wet roads focus on a single domain factor (road wetness), and the challenges associated with it is not the same as some navigating scenarios characterized by combinations of multiple factors, such as varying weather, diverse road conditions, and different car types, in addition to such wet surface. Our work aims to provide a novel parameter-efficient, sample-efficient RL adaptation structure, while leveraging pre-trained visual models, to tackle such complex realistic situations affected by multiple domain factors.
>
> [1] Duan, Jiafei, et al. "A survey of embodied ai: From simulators to research tasks." IEEE Transactions on Emerging Topics in Computational Intelligence 6.2 (2022): 230-244.
>
> [2] Kolve, Eric, et al. "Ai2-thor: An interactive 3d environment for visual ai." arXiv preprint arXiv:1712.05474 (2017).
>
> [3] Batra, Dhruv, et al. "Rearrangement: A challenge for embodied ai." arXiv preprint arXiv:2011.01975 (2020).
>
> [4] Weihs, Luca, et al. "Allenact: A framework for embodied ai research." arXiv preprint arXiv:2008.12760 (2020).
>
> [5] Dosovitskiy, Alexey, et al. "CARLA: An open urban driving simulator." Conference on robot learning. PMLR, 2017.
>
> [6] Todorov, Emanuel, Tom Erez, and Yuval Tassa. "Mujoco: A physics engine for model-based control." 2012 IEEE/RSJ international conference on intelligent robots and systems. IEEE, 2012.
>
> [7] Kempka, Michał, et al. "Vizdoom: A doom-based ai research platform for visual reinforcement learning." 2016 IEEE conference on computational intelligence and games (CIG). IEEE, 2016.
>
> [8] Petrenko, Aleksei, et al. "Megaverse: Simulating embodied agents at one million experiences per second." International Conference on Machine Learning. PMLR, 2021.

---

### Official Review · Reviewer_okL6 · 2023-07-11

**Soundness:** 3 good
**Presentation:** 3 good
**Contribution:** 3 good
**Rating:** 6
**Confidence:** 3

**Summary:**

The authors propose an approach for rapid policy adaptation to unseen visual observations for embodied agents. Their approach called CONPE (contrastive prompt ensemble) leverages contrastive learning to learn an ensemble of prompts for policy visual encoders that enables the policy to ultimately adjust to the domain shift. The results are demonstrated on multiple simulated environments, along with various ablations.

**Strengths:**

- The work addresses a well-motivated problem, which the authors explain well.
- The authors’ build up on advances in contrastive learning to develop a simple yet novel approach for policy adaption in embodied settings.
- The authors demonstrate their approach in multiple environments/tasks: AI2THOR, CARLA, and Metaworld.
- The authors also present extensive experiments, ablations, and analysis to provide a holistic understanding of their approach.

**Weaknesses:**

- The terminology used in the paper is confusing at places. The authors never define what they mean by a “prompt” or “zero-shot performance”. This is especially confusing in Sec. 3 where they explain the framework. In the overview in Sec.3.1 they talk about zero-shot policy deployment (L99). However, Sec.3.2 only talks about jointly learning the policy and the guided attention for the prompt ensemble (also main results tab.1 — see my comment in questions section on this). Overall, since the authors also use specialized data for contrastive learning of domain-invariant prompts to later enable policy adaptation (through joint learning of policy), I am wondering whether their approach can really be called zero-shot. Despite this, I do think their approach is useful for policy adaptation amongst domain shifts.  Hence, I recommend an acceptance. However, I encourage the authors to clarify their terminology and claims especially around “zero-shot”.
- The paper is also missing a limitation section. It is unclear where CONPE struggles. I encourage the authors to add one. What happens when a domain factor is varied beyond a degree present in the prompt learning data? e.g., a camera with much smaller view or more pan-tilt than present in the prompt training data. Lastly, I would also be interested in understanding how CONPE might work in real-world for instance, for robots.

**Questions:**

- It would be helpful to define source, seen and unseen target domains more concretely for Tab.1. I am a little confused between source and seen. Likewise, it would be useful to define what the authors mean by zero-shot for Tab.1 — aren’t a prompt pool and policy are jointly learnt for CONPE in Tab.1?
- L255: Unclear what ENS approach is. Please provide a 1-line description with a reference, if it is from an existing paper.
- Tab.5:  Shouldn’t E2E be the upper bound? Why does it have lower performance than CONPE?


**Limitations:**

(see also the weakness section)
- This might be out of scope for current work but the results are only shown on simulated environments. Would be great to see some discussion on how CONPE might work in real world.
- A limitation section is also missing.

---

> ### Author Rebuttal · Authors · 2023-08-09
>
> ### **Author response for Weakness 1. and Question 1. [prompt and zero-shot]**
> In Line 122 in Section 3.2, the term "prompt" is defined as a learnable vector appended to the input, with the intent of extracting domain-invariant knowledge from pre-trained visual encoders. This definition of prompt is commonly used in the context of pre-trained language and vision models, particularly in [1, 2, 3] where visual prompt-tuning is used similarly to our work.
>
> In our work, "zero-shot evaluation" pertains to the evaluation scenarios where a policy is evaluated in target domains, which were not encountered during its training. Accordingly, zero-shot evaluation occurs immediately, without any additional training.
>
> As clearly illustrated in Figure 2 of our manuscript, our training process consists of two training phases: (i) visual prompt-tuning (prompt-based contrastive learning) and (ii) policy's joint learning with the prompt ensemble (guided-attention-based prompt ensemble).
>
> In policy train, "source domain" is comprised of the domain factors that have seen in prompt learning.
> In our evaluation, the target domains are categorized as two different sets: "unseen target domains" and "seen target domain": the former denotes entirely new domains, which were not encountered during both (i) and (ii) training. In contrast, the latter denotes new domains, which are encountered during (i) but not during (ii).
> Such categorization of target domains aligns with the prior works in the application of visual domain RL adaptation such as [4, 5]. Note that the zero-shot performance for these target domains is represented in different columns in Table 1(a)-(c) of our manuscript.
>
> As formally defined in Section 2 of our manuscript, each domain is represented by a POMDP, and domain shifts are formulated by varied conditional observation probabilities. We also formulate each domain as an MDP (POMDP), defined by tuples of state, action, reward, and others, as same as [4, 5]. Domain examples in our datasets can be found in Tables 11-13 of Appendix.
>
> For clarity, we will update this explanation on zero-shot evaluation and target domain categorizations in our revised manuscript.
>
> ### **Author response for Weakness 2. and Limitations [missing limitation section]**
>
> We will highlight the limitations of our approach in the Conclusion Section, as recommended by the reviewer. Indeed, an inherent limitation of our ConPE framework is its architectural dependence on visual inputs. While this offers significant advantages in specific contexts, it might introduce challenges when the visual input is obscured and significantly varied as the reviewer mentioned.
>
> As briefly described as our future direction in the Conclusion Section of our manuscript, we consider semantic representations to tackle the limitations of visual input dependence. As a next step, we are keen on exploring the capabilities of multi-modal pre-trained models to further enhance embodied policy adaptation.
>
> As for adapting to field of view settings, we conduct several experiments in the AI2THOR environment, where we manipulate an agent's field of view to create either abnormally small (narrow observations) or excessively large (wide observations) views. Our ConPE framework shows slightly better performance than other baselines in both extreme cases, but its robustness appears to be variable.
>
> Similarly, for pan-tilt adjustments, we test with the Metaworld and CARLA environments. We observe performance inconsistency in Metaworld, with limited views surrounding a robotic arm. On the other hand, the CARLA environment demonstrates better resilience in ConPE's performance, likely due to the comprehensive camera settings capturing a wider range of relevant visual information.
>
> We also notice this inconsistency in performance could be attributed to the differences in image rendering quality and the volume of visual information available among the simulators. Further in-depth investigation is needed, as the reviewer queries, to achieve robustness across diverse simulators and environments.
>
> ### **Author response for Question 2. [ENS explanation]**
> ENS (ENSenble) refers to an ensemble method conducted on prompt embeddings, akin to the approach in [6], wherein prompts are integrated at the embedding-level.
> We further categorize this approach to ENS-UNI-AVG and ENS-WEI-AVG. In the former, the prompt embeddings are integrated using uniform average, while the latter, weighted averaging with attention is employed to integrate prompt embeddings.
>
> ### **Author response for Question 3. [performance score]**
> We notice such performance degradation in the end-to-end (E2E) setting, which is attributed to the necessity of fitting a larger number of parameters to the source domain. Updating a large model can lead to a decrease in overall generalization capability.
>
> This observation is consistent with findings reported in the RL literature [7], where the generalization capability tends to degrade after fine-tuning the entire model with a large number of parameters.
>
> [1] Jia, Menglin, et al. "Visual prompt tuning.", 2022.
>
> [2] Zhou, Kaiyang, et al. "Learning to prompt for vision-language models.", 2022.
>
> [3] Zhou, Kaiyang, et al. "Conditional prompt learning for vision-language models.", 2022.
>
> [4] Xing, Jinwei, et al. "Domain adaptation in reinforcement learning via latent unified state representation.", 2021.
>
> [5] Li, Dongfen, et al. "Domain adaptive state representation alignment for reinforcement learning.", 2022
>
> [6] Liu, Pengfei, et al. "Pre-train, prompt, and predict: A systematic survey of prompting methods in natural language processing.", 2023
>
> [7] Hakhamaneshi, Kourosh, et al. "Hierarchical few-shot imitation with skill transition models.", 2021

---

> > ### Comment · Reviewer_okL6 · 2023-08-22
> > **Thank you for the rebuttal**
> >
> > Thanks for the various clarifications. Despite some terminologies being well-known in a specific sub-field, the broader audience might not be aware of these. So I'd encourage the authors to ensure some of the clarifications they provided on their terminology is also captured in the main manuscript. I'd also like to see a limitation section hinting to the robustness experiments the authors mentioned (the experiments themselves can be in the appendix).
> >
> > I'll keep my rating.

---

> > > ### Author Response · Authors · 2023-08-22
> > >
> > > We appreciate your feedback. We acknowledge the importance of precise terminology in our manuscript. In response to your suggestion, we will update the descriptions about terms, and add 'Limitation' subsection and relevant experiments to briefly discuss the potential constraints of our study. We believe these modifications will enhance the clarity and comprehensiveness of our manuscript.

---

### Official Review · Reviewer_aYZP · 2023-07-12

**Soundness:** 3 good
**Presentation:** 3 good
**Contribution:** 3 good
**Rating:** 5
**Confidence:** 3

**Summary:**

The authors propose a method to address zero shot adaptation to novel visual domains. They do so by:
i) Learning a set of visual prompts via contrastive learning that encode specific domain invariant knowledge.
ii) Using a prompt ensemble based on embedding similarity scores (attention)

**Strengths:**

* The authors have incorporated the idea of image prompting to improve the model's capacity to adapt to novel environments in an effective manner.
* The experiments have been conducted on re implemented baselines from other works with all the hyperparameters specified in the appendix, which is deeply appreciated.
* The diagrams are a big help and very well done!

**Weaknesses:**

* I wish there were a deeper analysis on understanding/ interpreting the prompts better. Maybe something like visual analyses of images that tended to use similar prompt clusters thus seeing what invariant information the prompt encoded.
* I think the reader would benefit from a deeper understanding of the actual differences in the seen vs unseen domains, to get a sense of how similar or different they are.

**Questions:**


* Why was CLIP the choice of Visual encoder?
* Could you elaborate on the expert dataset used to do the prompt based contrastive learning?

---

> ### Author Rebuttal · Authors · 2023-08-09
>
> ### **Author response for Weakness 1.**
> In Figure 5(a)-1) of our manuscript, we visualize intra prompted embeddings from two distinct domains, both defined by different configurations of the same domain factor, to evaluate their alignment. The figure specifies that the prompted embeddings from these two domains align closely. For example, consider a scenario where the domain factor is "weather". The prompted embeddings might represent two different weather conditions, "clear" and "rainy", each presenting notably different visual representations. Even though these are two distinct weather scenarios, the embeddings align closely. This suggests that the prompt effectively captures the essence of weather as a domain factor. We include additonal examples in PDF as Global Author Rebuttal.
>
> Figure 5(a)-2) illustrates that each prompt distinctly clusters prompted embeddings corresponding to domains associated to its domain factor. If we consider multiple domain factors like "camera position", "weather", and "time of day", each prompt would cluster embeddings specific to its domain factor. For instance, one prompt might cluster all "rainy" observations, placing them nearer to each other compared to observations from different "camera position" or "time of day".
>
> We will include additional examples for intra and inter prompted embedding representations in Appendix of our revised manuscript.
> These visualizations provide insight into how these prompts capture domain-invariant knowledge with respect to a specific domain factor. They also demonstrate the efficacy of a similarity-based guidance during the ensemble process.
>
> Figure 5(b) illustrates the visualizations of attention weights during zero-shot adaptation.
> Each prompt receives varying attention weights across diverse domains, indicating ConPE's ability to dynamically adjust its focus using the perceived domain that is configured by multiple domain factors.
> For example, when the agent navigates in a "rainy" and "night" domain environment, the attention weights are likely higher for the prompt specifically trained for "weather" and "times of day" factors, as opposed to when navigating in a "high camera position" setting, where the prompt relative to "camera position" might be more emphasized.
>
> This demonstrates consistent assignments of attention weights to the prompted embeddings throughout the timesteps while adapting to the current domain.
>
> ### **Author response for Weakness 2.**
> In Tables 11-13 in Appendix, we present the configurations of our datasets including environment image samples and specific configurations used to set the values of domain factors.
>
> In the CARLA environment, we incorporate "weather conditions" and "times of day" as domain factors. The distinction between seen and unseen domains, combined by these two domain factors, aligns with the seen and unseen domain configurations presented in reference works [1, 2]. For example, "noon" and "sunny day" could be a seen domain, and "night" and "rainy weather" could be a unseen domain. As such, the differences between seen and unseen domains depend on configurations of domain factors in each simulator.
> It is also worth noting that our domain specification based on combinatorial configurations of multiple domain factors is scalable to include various domains, so other factors such as "camera positions", "camera field of view" and "different ranges of action magnitude" are seamlessly integrated, provided that they are supported and configurable in CARLA. We include domain examples in the PDF.
>
> We will include a detailed table to describe the differences of the seen and unseen target domains in Appendix of revised manuscript.
>
> ### **Author response for Question 1.**
> We use the CLIP model as our visual encoder since our primary intent is to leverage the capabilities of pre-trained visual encoders, which have seen significant advancements in the fields of computer vision, aiming to make the decoupled RL model structure more suited for zero-shot policy adaptation across domains.
> Our focus is on parameter-efficient approaches using visual prompt-tuning and ensembling for policy adaptation.
>
> Specifically, CLIP's Transformer architecture allows us to employ visual prompting (in a way that the prompt is incorporated into tokenized input embeddings) without much modification to the architecture.
>
> It is also worthwhile to note that our framework can accommodate other pre-trained encoders than CLIP, especially those based on the Transformer structure.
>
> In Table below, we conduct the experiment with another Transformer-based model (ViT [3]) learned on the ImageNet dataset. As shown, while ViT shows competitive performance for the source domains, its performance degrades significantly in zero-shot adaptation.
>
> We conjecture that this degradation of ViT is attributed to the relatively small size of training datasets (1,281,167 samples) used for visual encoder training. In contrast, CLIP is trained on the dataset of 400 million samples, thereby demonstrating robust generality.
>
> |Method|Source|Seen Target|Unseen Target|
> |-|-|-|-|
> |ViT[3] |$96.41 \%$|$27.34 \%$|$24.44 \%$|
> |ConPE  |$96.11 \%$|$87.50 \%$|$73.21 \%$|
>
> ### **Author response for Question 2.**
> The statistics of our expert datasets used for prompt learning are detailed in Tables 11-13 in Appendix.
> Each dataset contains expert demonstrations from multiple domains, where each domain is determined by varying a specific domain factor.
>
> We use 28,464 samples for AI2THOR, 4,700 samples for Metaworld, and 73,900 samples for CARLA, and regarding to the experts, we devise rule-based policies tailored to each environment.
>
> [1] Xing et al. "Domain adaptation in reinforcement learning via latent unified state representation.", 2021.
>
> [2] Li et al. "Domain adaptive state representation alignment for reinforcement learning.", 2022
>
> [3] Dosovitskiy et al. "An image is worth 16x16 words: Transformers for image recognition at scale.", 2020

---

### Official Review · Reviewer_TuT5 · 2023-07-13

**Soundness:** 3 good
**Presentation:** 3 good
**Contribution:** 3 good
**Rating:** 6
**Confidence:** 4

**Summary:**

This paper presents a method for improving zero-shot domain adaptation for Embodied AI agents by learning a task-specific state representation.

Specifically, $n$ visual prompts are learned by fine-tuning the agent's visual encoder (i.e. CLIP) contrastively on observations. Each visual prompt is specific to a "domain factor" (such as camera position or illumination). Observations are positively paired if their state is the same (but the observations themselves may differ depending on domain factors).

Then, an ensembling technique is used to compute a task-specific state representation. This is an average of the prompt encodings, weighted based on their cosine similarities to the observation encoding. This is used to optimize the policy using RL.

**Strengths:**

- The method presented in this paper is relatively simple and appears to provide significant gains over SOTA for both source and (seen/unseen) target domains on several benchmarks (AI2-THOR tasks, egocentric-Metaworld, CARLA).

- To the best of my knowledge, visual prompting has not previously been explored in Embodied AI. This paper also emphasizes learning visual variations that are specific to domain factors (and shows that this helps generalize to unseen domains).

- This paper includes several ablations, e.g. for prompt ensemble scalability and ensembling methods.

- Zero-shot domain adaptation is a necessary and under-explored capability in Embodied AI, as it has many downstream applications (e.g. Sim2Real for robotics).

**Weaknesses:**

- It's a bit hard to interpret the sample efficiency experiment in Fig. 4. For the seen target domain, it's true that the performance is higher for ConPE than EmbCLIP at 0.5M steps, before either method has converged (approx. at 1M steps). But it's a bit sparse, as success is only measured every 0.5M steps (and the performance for ConPE is always higher than EmbCLIP anyway). Something somewhat similar could be said for the unseen target domain. Also, what benchmark does this correspond to? If this is averaged over several benchmarks, these should probably be split into different plots.

- I believe the details related to the domain factors could be clarified. This seems to be an important component of the methodology and several things are unclear: what is the full list of domain factors (and do these vary per environment?), how were these domain factors chosen, what is the variance per domain factor within one environment (are these discrete or continuous values? are they uniformly sampled per observation?), and how were these subsampled for the experiment in Table 3. Most importantly: how does visual prompting perform without any domain factors? I.e. what are the baseline metrics if you fine-tune a single visual prompt for each environment?
  - These points are particularly important for the scientific contribution of this work.

**Questions:**

- It is said that $T_\phi$ is "fine-tuned" but I believe this must be frozen and only the visual prompts are fine-tuned. Is that correct? Should be clarified if so.

- Could the authors please clarify whether they use RoboTHOR or iTHOR for the AI2-THOR benchmarks? E.g. the ObjectNav task is available for both environments.

**Limitations:**

As far as I understand, this method may not be generalizable between simulators/environments, because the domain factors may not be compatible between these. Future work will be necessary to help improve such generalization -- fine-tuning the visual encoder offline is likely a valid approach, but factor-specific visual prompting or ensembling is not.

---

> ### Author Rebuttal · Authors · 2023-08-09
>
> ### **Author response for Weakness 1.**
> |Method|0.5 M|0.6 M|0.7 M|0.8 M|0.9 M|1.0 M|
> |-|-|-|-|-|-|-|
> |EmbCLIP    |$54.38$|$60.35$|$60.00$|$66.00$|$66.38$|$70.72$|
> |ATC        |$56.61$|$54.30$|$54.12$|$52.54$|$61.61$|$60.45$|
> |ConPE      |$64.52$|$68.16$|$69.91$|$71.58$|$71.50$|$75.53$|
>
> We appreciate this reviewer's valuable feedback on the sample efficiency in Figure 4 of our manuscript.
>
> Figure 4 shows the sample efficiency for the Object Navigation task in AI2THOR.
> To clarify the sample efficiency, we measure the policy's performance at each 0.1 million timesteps during 1 million timesteps, as illustrated in the tables above.
>
> For the seen target domains, we observe that ConPE requires, on average, 30.0\% fewer samples to achieve approximately 70.0\% success rate, compared to the most comparative baseline. These results indicate that the framework is the most sample efficient for zero-shot adaptation to seen target domains.
>
> We will update Figure 4 in our revised manuscript, showing the detailed learning progress with more plots during the initial stages before 1.5 million timesteps for both seen and unseen target domains.
>
> ### **Author response for Weakness 2-1**: [list of domain factors, how chosen, what is the variance (discrete or continuous values, sampling)]
> The examples of domain factors for each environment (i.e., AI2THOR, Metaworld, CARLA) are provided in Appendix D. We will include the domain factor list (i.e., Camera field of views, stride length, rotation degrees, and illuminations in AI2THOR; Camera positions, gravity, wind speeds, and illuminations in Metaworld; Camera positions, camera field of views, weather conditions, times of day, and different ranges of action magnitude in CARLA.) in our revised manuscript.
>
> We have chosen these domain factors based on a couple of considerations: how significantly they affect domain shifts in relation to the agent's performance, and how effectively we can create different domain data for each domain factor through diverse configurations of the simulators we use. We also consider previous studies in embodied RL [1, 2, 3] in which domain factors like camera position, camera FoV, and stride length were used.
>
> Each domain factor can be represented as either discrete or continuous values, depending on its inherent nature. For example, we treat weather conditions as a discrete factor, which can be classified as either clear, rainy, cloudy, or other; Conversely, we represent brightness as a continuous factor, with a range extending from $0.0$ to $1.0$.
>
> Regarding sampling methods, these domain factors are individually drawn from a uniform distribution to produce combinatorial domain variations within the simulation environment.
>
> We will detail these domain factors including the sampling method in Appendix of our revised manuscript.
>
> ### **Author response for Weakness 2-2**: [experiment in Table 3]
> In Table 3 of our manuscript, $n$ represents how many prompts we use for training domain factors within the AI2THOR object navigation environment. In this experiment, we use total 18 different domain factors in AI2THOR to show the scalability of our framework in terms of multiple visual prompts.
>
> For example, when $n=5$, the domain factors used include illuminations, FoV, stride length, rotate degree, and look up/down degree, thus requiring the respective 5 individual prompts in our ensemble structure.
>
> ### **Author response for Weakness 2-3**: [without domain factors, with a single visual prompt]
> For a specific domain in the environment, it is feasible to fine-tune the corresponding visual prompt for policy adaptation, aiming to improve the performance of the source domain. However, it is expected that the tuned visual prompt easily overfits to the source domain, that is trained.
>
> As our objective is to ensure robust zero-shot performance for unseen target domains other than the source domain, this situation is not desirable. The zero-shot performance can significantly degrade when overfitted. Thus, we explore an ensemble structure of multiple visual prompts, each encapsulating knowledge pertinent to its respective domain factor.
>
> In fact, if we showed the case of $n=1$ in Table 3, it would correspond to the single prompt case. Given the low performance result by the other case of $n=2$ in the table, we expect a relatively low performance by the case of $n=1$ for seen and unseen target domains. We will confirm this in our revised manuscript.
>
> We will also expand this comparison including the single prompt optimized to the source domain as well as the domain-randomized single prompt in our revised manuscript.
>
> ### **Author response for Question 1.**
> As noted by the reviewer, the visual encoder $\mathcal{T}_{\phi}$ remains frozen and only the visual prompts are updated, same as other visual prompt-tuning approaches [4, 5, 6].
>
> We will use the term "visual prompt-tuning" than "fine-tuning" in our revised manuscript. This term more accurately conveys our intent, tuning only the visual prompts, as opposed to the encoder parameters.
>
> ### **Author response for Question 2.**
> We use iTHOR for the AI2-THOR benchmarks, because iTHOR involves more diverse object types (e.g., fridge, Window, Stove burner) and tasks (e.g., Object Goal Navigation, Point Goal Navigation, Room Rearrangement, Image Goal Navigation), compared to RoboTHOR.
>
> [1] Chattopadhyay, Prithvijit, et al. "Robustnav: Towards benchmarking robustness in embodied navigation.", 2021.
>
> [2] Shah, Dhruv, et al. "Gnm: A general navigation model to drive any robot.", 2023.
>
> [3] Julian, Ryan, et al. "Efficient adaptation for end-to-end vision-based robotic manipulation.", 2020
>
> [4] Jia, Menglin, et al. "Visual prompt tuning." , 2022
>
> [5] Sohn, Kihyuk, et al. "Visual prompt tuning for generative transfer learning.", 2023.
>
> [6] Peng, Xiangyu, et al. "Model ensemble instead of prompt fusion: a sample-specific knowledge transfer method for few-shot prompt tuning.", 2022

---

> > ### Comment · Reviewer_TuT5 · 2023-08-20
> >
> > Thanks to the authors for responding to my review. I have now looked over all the reviews and their corresponding responses. I am satisfied with the responses provided by the authors and appreciate all their clarifications. In particular, my biggest initial confusion was around the selection of domain factors, but I feel that that has been explained now --- and I hope the authors will update their manuscript accordingly.
> >
> > I intend to retain my score of Weak Accept and am pleased to see that (on average) the reviewers are similarly inclined to accept this paper.

---

> > > ### Author Response · Authors · 2023-08-21
> > >
> > > As the end of the author-reviewer discussion period approaches, we would like to express our gratitude once again for your insightful feedback during the review process. We hope that our responses addressed your remaining concerns. Based on the review, we will incorporate the suggested feedback for clarity into our manuscript.
> > >
> > > If you have additional questions or need further information, please don't hesitate to discuss, and we'll gladly provide further clarity.

---

### Official Review · Reviewer_QsRA · 2023-07-27

**Soundness:** 3 good
**Presentation:** 1 poor
**Contribution:** 2 fair
**Rating:** 5
**Confidence:** 4

**Summary:**

This paper presents a novel framework for “efficient policy adaptation”, i.e. adapting learned policies to changes in the domain (e.g., lighting, camera height etc.), using a very small amount of in-domain data (few-shot and zero-shot). The proposed approach comprises learning “visual prompts” for modeling each different kind of variation in the domain, and using an attention head over these prompts to learn policies that are invariant to these factors. Empirical analysis in three different settings (AI2THOR, Metaworld and CARLA) show that the proposed method outperforms other representation learning alternatives.

**Strengths:**

The paper studies a problem that is central to the sim2real generalization of robotic policies trained in simulation: how well do these policies generalize to varying conditions in the domain that are out of the agents control. The proposed approach, while having a lot of moving parts, seems to be extremely data-efficient (~10 expert episodes and small amount of diversity used), which can make a strong case for real-world applications. I also found the ablation analysis to be thorough and convincing regarding the design decisions made.

**Weaknesses:**

- [Method Presentation, Reproducibility] My biggest concern with the paper in it’s current form is the poor presentation of method details and reproducibility. I appreciate that the authors have provided code in the supplemental material, which will really help reproduce the results and build on them, but as a research article of novel, scholarly work, the paper is extremely hard to parse and missing on several key details. Updating these and clarifying the integral components of the paper better would be imperative to the impact of the paper. Some questions:
	- In Section 3.2, it is not clear what the learned parameters are: only the prompts $p^v$, or both prompts and $\mathcal{T}_\phi$. The objective only says $p^v$, but the text says “we fine-tune $\mathcal{T}_\phi$ by learning a visual prompt…” This is _really_ important to understanding why the method works: if T is indeed being finetuned, how is this standardized across the tasks/domains? Why would $p^v$ learn anything useful or interpretable at all?
	- I don’t think it is ever mentioned what specific pre-trained representation $\mathcal{T}$ actually is. At some point, the authors say it is CLIP but it is not clear how CLIP is used. CLIP consists of two encoders (text and visual): which of these is $\mathcal{T}$? It also seems to be overloaded as something with one or two arguments, and while this is somewhat understandable in context, this should be clarified for correctness.
	- “each visual prompt in $p^v$ encapsulates a specific domain-invariant knowledge” is a somewhat strong statement. Is there any reason this is interpretably something domain-invariant? Empirical/illustrative insights would really be helpful here, or even a qualitative discussion.
	- [L141-143] This statement is very convoluted and seems unclear. Can the authors very concretely define what they mean by “task” and “domain”? Maybe this is well-understood by a sub-community, but certainly seems vague to me in the context of a representation learning paper. "As G is optimized by a given RL task objective in the source domains (in line 12), the resulting Z tends to be task-specific, while Z is also domain-invariant by the ensemble of contrastively learned visual prompts based on G with respect to various domain factors.”

- [Prior work] In the introduction, the authors make some rather bold claims that I think should be qualified. In [L27-30], the authors say “It was rarely investigated how to optimize…, to ensure the zero-shot capability of embodied agents”. Two thoughts here:
	- It should be clarified if this statement is regarding zero-shot generalization of visual encoders or entire policies. This has indeed been studied by many works: both for encoders [5] and policies, most recently in the navigation/embodied AI settings by [1, 2]. It would be nice to acknowledge some of this prior work and position the paper more carefully in their context (not saying there are no differences or it reduces the contributions of this paper).
	- And for what it’s worth, the current paper does not really address this. The closest reported results analyze how policies trained with a small set of domain properties generalize to slightly different properties.

- This trend continues with statements like “egocentric visual domain changes experienced by embodied agents due to different camera positions”. This has indeed been studied for visual encoders [5, 6] trained on diverse data such as Ego4D, as well as for policies [1, 2, 3] (both these papers specifically studied the case of different camera positions and FoV, and even different embodiments etc.).

- [Task & eval Description] Some of the lack of key information seeps into the evaluations as well. Separating it for better readability. I was unable to interpret the results without getting more clarity on what the experiment setup is and what is being reported. Please see some specific concerns below:
	- What do tasks/domains etc. mean specifically for each of the environments considered in the paper? Some details seem to be in the appendix, but even a brief discussion for completeness in the main paper with pointers to appendix should suffice.
	- I don’t think the specific task in CARLA is defined anywhere, even in the appendix.
	- What does “zero shot” mean in Section 4.1? Is it trained in the same environment for the same task but with different lighting conditions/camera parameters etc.? Or is everything different? What is the exact experiment? The text as well as the captions are uninformative about this (e.g., “The Seen Target column presents the performance for the seen target domains, and the Unseen Target column presents the performance for the unseen target domains.”) I suspect that the words “zero-shot performance” are too vague and broad and mis-communicate some of the findings. I would like to see some clarity on this before I can interpret the results.
	- The caption in Fig 5 is pretty hard to parse, could the authors clarify what is being visualized in the left image and why it indicates "domain-invariant knowledge” ?
	- What is the “pretrained policy” in Section 4.2 actually pretrained on? It seems like individual policies are used for each of the different tasks in each environment, so what exactly is being “pretrained”? It’s really not clear what the distinction between pretrain/prior results are, or what tasks/domains mean and what the exact experiment is.
	- In the experiments, I see that the policy training used ~10 episodes per task. How much data was used for training the visual prompts, though? Seems like that would be quite high (which is fine since this is likely done procedurally in sim).

- A broader question regarding the setup of the experiments to me is how much of an improvement a sophisticated scheme like the one proposed over something more naive, like (i) learning a parametric policy that simply conditions on the different factors of variation [3], or (ii) training a joint policy on data from multiple domains across all these parameters, akin to domain randomization, [1, 2] so the factors of variation can be learned to be ignored. I imagine either of these “simpler” methods would require more data, but given this is a study in sim where data is free, does the proposed method help in certain ways? An analysis of this could be really important.

[1] Putta et al., “Emergence of Implicit System Identification via Embodiment Randomization”, 2023

[2] Shah et al., “GNM: A General Navigation Model to Drive Any Robot”, 2022

[3] Hirose et al., “ExAug: Robot-Conditioned Navigation Policies via Geometric Experience Augmentation”, 2022

[5] Majumdar et al., “Where are we in the search for an Artificial Visual Cortex for Embodied Intelligence?”, 2023

[6] Nair et al., “R3M: A Universal Visual Representation for Robot Manipulation”, 2022

**Questions:**

Please see the concerns and questions raised in the detailed weaknesses section above.

**Limitations:**

**I did not find limitations** or societal impact sections (the latter is likely not required) in the paper.

---

> ### Author Rebuttal · Authors · 2023-08-10
>
> ### **Weakness 1-1.**
> In Sec. 3.2, when using the phrase "fine-tune $\mathcal{T}_{\phi}$", it denotes the tuning process of visual prompts, a process we term visual prompt-tuning. This does not encompass parameter updating $\phi$ in CLIP visual encoder $\mathcal{T}$. Via the visual prompt-tuning, the encoder's parameters $\phi$ remain frozen, precluding  modification.
> To eliminate any misunderstanding, we will substitute fine-tuning with prompt-tuning.
> ### **Weakness 1-2.**
> With CLIP visual encoder, we adopt the prompt-based contrastive learning to extract features pertinent a specific factor. We construct an ensemble of the visual prompts to manage domain shifts brought about by a combination of these factors. By learning each visual prompt with a few demos (while CLIP encoder is frozen), we can encapsulate knowledge pertinent to a specific factor within a prompt.
> We also use the CLIP text encoder for Semantic Regularized Data Augmentation in Table 6.
> ### **Weakness 1-3.**
> To mitigate discrepancies between domains, we incorporate contrastive learning into our adaptation approach. It is consistent with [7,8], for which contrastive learning serves for aligning the representations from domains, enabling cross-domain transfer of knowledge.
> We empirically prove that our framework generalizes for diverse domain scenarios in Sec. 4.1 and Appendix F. Those include 1) Alignment visualization for prompted embeddings of different domains, 2) Robustness of the ensemble representation across different domains w.r.t a specific domain factor, and 3) Interpretability of attention weights prompt ensemble.
> ### **Weakness 1-4.**
> Our work aligns with visual RL that "task'" refers to a specific objective of an agent (e.g., Object Goal Navigation), and determined by rewards.
> A "domain" pertains to the visual observation space and observation probability (e.g., domains with a high camera position, the agent receives high position camera views.); each is characterized by a combination of factors, as in Lines 86-94.
> We include the definition table in PDF as Global Author Rebuttal.
> ### **Weakness 2-1.**
> We appreciate the reviewer's valuable feedback on related works in robotics.
> We will revise the Introduction and Related Work to articulate the position of our works with references [1, 2, 3, 5, 6].
> Our work distinguishes itself from the prior works in that the visual encoder's prompts are only updated for policy adaptation, and it provides a novel parameter-efficient, sample-efficient RL adaptation structure, while leveraging pre-trained visual models.
> ### **Weakness 2-2.**
> Generalizing for each individual domain factor can provide a basic understanding, but we presume it doesn't necessarily prepare an agent for intricate multi-factorial situations; e.g., mastering driving on wet roads focuses on a single factor (road wetness), and the challenges associated with it is not the same as navigating scenarios characterized by combinations of multiple factors: weather, road conditions, and car types.
> ### **Weakness 4-3.**
> "zero-shot" pertains to the evaluation scenarios where a policy is evaluated in target domains, not encountered during the training.
> As in Figure 2, our training consists of two phases: (i) visual prompt-tuning (prompt-based contrastive learning) and (ii) policy's joint learning with the prompt ensemble (guided-attention-based prompt ensemble).
> In evaluation, the target domains are either "unseen target domains" or "seen target domain": the former denotes entirely new domains, which were not encountered during both (i) and (ii) training. In contrast, the latter denotes new domains, which are encountered during (i) but not during (ii).
> ### **Weakness 4-4.**
> In Figure 5(a)-1), we visualize prompted embeddings of two distinct domains, both defined by different configurations of the same factor, to evaluate the alignment; prompted embeddings from these two domains align closely.
> Figure 5(a)-2) illustrates that each prompt distinctly clusters prompted embeddings corresponding to domains associated to its domain factor.
> Both visualizations provide insight into how these prompts capture domain-invariant knowledge with respect to a specific domain factor.
> Figure 5(b) illustrates the visualizations of attention weights during zero-shot adaptation. Each prompt receives varying attention weights across diverse domains, indicating ConPE's ability to dynamically adjust its focus using the perceived domain that is configured by multiple domain factors.
> ### **Weakness 4-5.**
> In Sec. 4.2, "pre-trained" denotes a policy fully trained in the source domains for a specific RL task.
> (i.e., associated with a reward function).
> For a pre-trained policy, the model architecture for training it employs the CLIP visual encoder without prompts. This is different from the other cases of zero-shot performance, in Sec. 4.1, where a policy is jointly learned with the prompt ensemble through an RL objective.
> ConPE supports two methods: joint learning in Sec. 4.1 and tuning of the prompt ensemble with a pre-trained policy in Sec. 4.2.
> ### **Weakness 4-6.**
> The datasets comprise 28,464 samples for AI2THOR, 4,700 samples for Metaworld, and 73,900 samples for CARLA, respectively (details in Table 11-13 of Appendix.).
> ### **Weakness 5.**
> Regarding (i): as shown in [3], the policy conditioned on known specific parameters is quite intuitively structured in the perspective of multi-task learning, where a distinctly parameterized condition can be used as input to infer task-specific contexts of multi-tasks.
> Unlike this parametric policy, we do not assume any known configurations of domain factors for target domains. This evaluation assumption is consistent with zero-shot policy adaptation in target domains.
> Regarding (ii): we will include test on domain randomized encoders in revised manuscript.
>
> [7] Kang et al. "Contrastive adaptation network for unsupervised domain adaptation.", 2019
>
> [8] Thota et al. "Contrastive domain adaptation.", 2021

---

### Author Rebuttal · Authors · 2023-08-10

We attached global author rebuttal. Please refer to it.

---

> ### Comment · Area_Chair_a8Nh · 2023-08-21
> **Thanks for detailed rebuttal**
>
> Thanks to the authors for individual responses to the reviewers that span readability clarifications, training data and procedure, details about prompting, discussion of limitations, etc. The committee will discuss further and take these into consideration.

---

> > ### Author Response · Authors · 2023-08-21
> >
> > We sincerely appreciate your careful consideration.
> > Throughout the discussion period, we are committed to offering clear and comprehensive responses to the reviewers' queries. It is our utmost priority to furnish any and all information the reviewers may require.
> >
> > Once again, we deeply value your meticulous attention to our work.

---

### Decision · Program_Chairs · 2023-09-21

**Decision:**

Accept (poster)

**Comment:**

This paper focussed on zero-shot domain adaptation to unseen visual observations for embodied agents. The proposed approach CONPE (contrastive prompt ensemble) fine-tunes the observation/visual encoder contrastively to tackle domain gaps. An ensemble over this  helps with representations for specific tasks, demonstrated across AI2THOR, Metaworld, and CARLA simulators, widely used for training embodied perception-control agents.

This submission received six reviews that unanimously support acceptance. Key strengths that most reviewers agree on are (1) simple and effective approach tackling generalization via zero-shot adaptation, (2) extensive empirical evidence across several simulators, tasks, and baselines, (3) novelty around marrying visual prompting with embodied actor policies, (4) clear presentation via helpful visuals and analysis. Two reviewers updated reviews (including one rating update) after the author response period, which were also incorporated in the decision-making.

The AC concurs with the unanimous decision of the reviewers and recommends acceptance. The authors are encouraged to utilize their constructive and detailed discussion with reviewers towards further raising the quality and impact of their work.